

# Temporal changes in the most effective pollinator of a bromeliad pollinated by bees and hummingbirds

Roberta Luisa Barbosa Leal[1], Marina Muniz Moreira[1,2],
Alessandra Ribeiro Pinto[3], Júlia de Oliveira Ferreira[1],
Miguel Rodriguez-Girones[4] and Leandro Freitas[1]

[1] Jardim Botânico do Rio de Janeiro, Rio de Janeiro, Brazil
[2] Centro de Ciências Agrárias, Universidade Federal do Espírito Santo, Alegre, Espírito Santo, Brazil
[3] Programa de Pós-Graduação em Ecologia, Universidade Federal do Rio de Janeiro, Rio de Janeiro, Brazil
[4] Estación Experimental de Zonas Áridas, Almería, Spain

## ABSTRACT

A generalist pollination system may be characterized through the interaction of a plant species with two or more functional groups of pollinators. The spatiotemporal variation of the most effective pollinator is the factor most frequently advocated to explain the emergence and maintenance of generalist pollination systems. There are few studies merging variation in floral visitor assemblages and the efficacy of pollination by different functional groups. Thus, there are gaps in our knowledge about the variation in time of pollinator efficacy and frequency of generalist species. In this study, we evaluated the pollination efficacy of the floral visitors of *Edmundoa lindenii* (Bromeliaceae) and their frequency of visits across four reproductive events. We analyzed the frequency of the three groups of floral visitors (large bees, small bees, and hummingbirds) through focal observations in the reproductive events of 2015, 2016, 2017, and 2018. We evaluated the pollination efficacy (fecundity after one visit) through selective exposure treatments and the breeding system by manual pollinations. We tested if the reproductive success after natural pollination varied between the reproductive events and also calculated the pollen limitation index. *E. lindenii* is a self-incompatible and parthenocarpic species, requiring the action of pollinators for sexual reproduction. Hummingbirds had higher efficacy than large bees and small bees acted only as pollen larcenists. The relative frequency of the groups of floral visitors varied between the reproductive events. Pollen limitation has occurred only in the reproductive event of 2017, when visits by hummingbirds were scarce and reproductive success after natural pollination was the lowest. We conclude that hummingbirds and large bees were the main and the secondary pollinators of *E. lindenii*, respectively, and that temporal variations in the pollinator assemblages had effects on its reproductive success. Despite their lower pollination efficacy, large bees ensured seed set when hummingbirds failed. Thus, we provide evidence that variable pollination environments may favor generalization, even under differential effectiveness of pollinator groups if secondary pollinators provide reproductive assurance.

Corresponding author
Leandro Freitas,
lfreitas.jbot@gmail.com

Pollen limitation, Floral reflectance, Nectar, Bromeliaceae, Self-incompatibility, Tropical forest,
Pollen robber

## INTRODUCTION

In most plants, flowers are visited by a diverse assemblage of animals, which characterizes generalist pollination systems (*Waser et al., 1996*). Based on the behavior and morphophysiological traits, floral visitors can be arranged in different functional groups of pollinators, which may differ in their contribution to the plant reproductive success (*Fenster et al., 2004*). This difference in contribution occurs as these pollinators can vary in frequency of visits and ability to transfer pollen (*Shuttleworth & Johnson, 2008*; *Ollerton, 2017*). Studies encompassing generalist pollination systems mainly report floral visitor assemblages and visitation rates (*Thompson, 2001*; *Freitas & Sazima, 2006*; *Scrok & Varassin, 2011*) despite that not all visitors are actual pollinators (*Armbruster, Fenster & Dudash, 2000*; *Ollerton, 2017*). Therefore, studies that evaluate the pollination efficacy (for instance, measuring seed set after one visit; *Freitas, 2013*) of different functional groups of pollinators are necessary to better understand pollinators' relevance on plants with a generalist pollination system.

Based on the principle of "the most effective pollinator," a plant having more than one functional group of pollinators could be interpreted as an intermediary stage in the shift from one specialized pollinator to another (*Stebbins, 1970*). An alternative explanation is that generalist systems are not transient and may be favored in certain scenarios, for example, under unpredictable pollination environments (*Herrera, 1988*, *1996*; *Waser et al., 1996*; *Armbruster, Fenster & Dudash, 2000*; *Gómez & Zamora, 2006*; *Ollerton et al., 2007*). Accordingly, the spatiotemporal variation of the most effective pollinator is the factor most frequently advocated to explain the emergence and maintenance of generalist pollination systems. Several studies have explored temporal variation in composition and frequency of floral visitors (*Fenster & Dudash, 2001*; *Ivey, Martinez & Wyatt, 2003*; *Zych et al., 2018*). Other studies have quantified the effectiveness (i.e., the product of efficacy times visitation rate, after *Herrera (1987)* and *Freitas (2013)*) of different pollinator species or functional groups (*Amorim, Galetto & Sazima, 2013*; *Salas-Arcos, Lara & Ornelas, 2017*). However, information about floral visitor assemblages and their effectiveness on pollination over multiple reproductive events is restricted to a few systems (*Herrera, 1987*, *1988*; *Larsson, 2005*; *Wiggam & Ferguson, 2005*). Thus, there are gaps in knowledge about the variation of pollinators in generalist plants and their effectiveness over time.

Generalized pollination systems have ecological and evolutionary dimensions (*Armbruster, Fenster & Dudash, 2000*), therefore the effect of several pollinators in the process of evolutionary generalization depends on the selective pressures exerted by those floral visitors. In this sense, differences in pollination efficacy (sensu *Freitas (2013)*) among functional groups may be enhanced if the variations in the pollination environment affect the plant reproductive success. For instance, pollen limitation (PL), the lower fruit
and/or seed production due to inadequate pollen receipt, is widespread in angiosperms (*Ashman et al., 2004*; *Knight et al., 2005*) and similarly to the pollinator effectiveness, its magnitude varies at several scales (*Bennett et al., 2018*). However, how temporal variations in the pollination environment and PL levels are related is a fundamental but poorly understood aspect to a better understanding of the mechanisms that lead to the maintenance of generalized pollination systems (*Koski et al., 2018*).

Bromeliaceae is the largest family of predominantly Neotropical angiosperms (*Givnish, 2017*) and hummingbirds are the main group of pollinators of them (*Benzing, 2000*). However, several species are known to have mixed pollination systems, for example, involving hummingbirds and bees or bats (*Givnish et al., 2014*). *Edmundoa lindenii* (Regel) Leme bears flowers that are visited by both hummingbirds and bees in montane tropical forests, however we do not know the role of those groups on the plant sexual reproduction over time. Here, we evaluated the efficacy of three groups of floral visitors and their frequency of visits over four reproductive events. In particular, we addressed the following questions: (1) Do hummingbirds, large bees, and small stingless bees pollinate this species, considering the high divergence of traits between them? (2) Is the pollination efficacy of hummingbirds higher than large bees, since hummingbirds pollinate most species of Bromeliaceae? (3) Are the relative frequencies of floral visits by each group equivalent over four reproductive events, since clear environmental variations were not noted in those years? (4) However, if they are not equivalent, are those variations in frequency related to the reproductive success in natural conditions and the occurrence of pollen limitation?

## MATERIALS AND METHODS

### Study site and species

This study was conducted in an area covered by montane Atlantic Forest, located in the Serra dos Órgãos National Park (PARNASO), Rio de Janeiro state, Brazil (22° 52′–22° 54′ S and 42° 09′–45° 06′ W, ca. 960 m a.s.l.) among four reproductive events (from 2014 to 2018). The average annual rainfall at the study site is 2,436 mm, with the rainiest period between December and March and colder and drier months from June to August. The mean annual temperature is 18.6 °C, with minimum and maximum monthly temperatures of 13.7 °C and 22.9 °C (climate data for 2015 to 2018 from the meteorological station located inside the PARNASO). The field research reported here was performed using the required permit (SISBIO Nos. 34882 and 432793).

*Edmundoa lindenii* (Regel) Leme (Bromeliaceae-Bromelioideae) is a terrestrial, saxicolous, or epiphyte herb, endemic to the Atlantic Forest in south and southeastern Brazil (*Martinelli et al., 2008*; *Zappi et al., 2015*). In the study area, this species blooms between December and February and produces fruits between March and April; its flowers are visited by bees and hummingbirds (R.L.B. Leal et al., 2015, personal observations).

### Floral biology

We measured inflorescences of *E. lindenii* (*n* = 16 individuals) directly in the field with a measuring tap, considering the following traits: scape length, inflorescence diameter and

bract length. Flowers ($n = 73$) from 28 individuals were collected in the field, stored in 70% alcohol and measured in the laboratory with a digital caliper considering the following structures: corolla tube length (i.e., from septal nectary to the opening of the corolla) and the width of the corolla tube opening. We counted the number of ovules in 25 flowers ($n = 15$ individuals).

To analyze the color quantitatively, we measured the spectral reflectance of petals, sepals and bracts. For this, 12 flowers ($n = 6$ individuals) were collected in the field, stored in thermal bags containing moist paper and brought to the laboratory, where they were immediately measured (*Lunau et al., 2011*). We measured the reflectance using an USB2000 spectrophotometer (OceanOptics, Inc., Dunedin, FL, USA) coupled with a deuterium–halogen light source (DH-2000; OceanOptics, Inc., Ostfildern, Germany), with a light emission range between 215 and 1,700 nm. We took all reflectance measurements at a 45° angle in relation to the plant structure and we used barium sulfate as the white standard and black paper as the black standard (*Chittka & Kevan, 2005*).

We used the logarithm version of the receptor noise-limited model (*Vorobyev et al., 2001*) to compare the colors of the petals, sepals and bracts. Chromatic distances were calculated according to the trichromatic formulation for bees and the tetrachromatic formulation for hummingbirds (*Vorobyev & Osorio, 1998*). We modeled spectral sensitivity curves using data from *Sephanoides sephaniodes* (*Herrera et al., 2008*) to estimate hummingbird color distances and from *Bombus terrestris* for bees (*Telles & Rodríguez-Gironés, 2015*). In all cases, we used standard daylight illumination (D65—*Wyszecki & Stiles, 1982*). Using these models, we determined the spectral location of each structure in a color space for each pollinator.

The distance between two points in a color space provides an approximation of the perceived color difference (*Endler & Mielke, 2005*). We evaluated color distances between sepals, petals and bracts. Using the receptor noise-limited model, we estimated that two colors were discriminable if their distance was greater than 0.27 units for bees (*Telles & Rodríguez-Gironés, 2015*) and 1.0 for hummingbirds (*Vorobyev et al., 1998*). For representation, we also calculated the color loci of the flower colors in the respective color space models: the color hexagon for bees (*Chittka, 1992*) and the color tetrahedron for hummingbirds (*Vorobyev et al., 1998*).

## Nectar

We measured the nectar volume in flowers previously bagged in bud stage, with a graduated microliter syringe (Hamilton, NV, USA) and the concentration with hand-held refractometer (Bellingham + Stanley Eclipse, UK). To evaluate nectar production during anthesis, without the effect of nectar removal, 36 flowers ($n = 10$ individuals) previously bagged at the bud stage were measured once after anthesis onset. In total, we performed measurements at four different times of the day: 7:00 AM ($n = 10$ flowers); 8:30 AM ($n = 10$); 10:00 AM ($n = 10$); and 11:30 AM ($n = 6$). To evaluate if nectar removal stimulates its secretion, 24 flowers were submitted to four treatments ($n = 6$ individuals, four flowers per individual—one flower per treatment per individual): R = nectar was measured once at 11:00 AM; R1 = nectar was measured twice (at 10:00 AM and 11:30 AM);
R2 = nectar was measured three times (at 8:30 AM, 10:00 AM and 11:30 AM); and R3 nectar was measured four times (at 7:00 AM, 8:30 AM, 10:00 AM and 11:30 AM). Thus, for the treatments R1, R2 and R3, nectar was remeasured on the same flowers. We calculated the total amount of sugar (mg) per flower by multiplying nectar volume (µL) by its corrected concentration (mg/µL) according to *Dafni, Kevan & Husband (2005)*.

## Breeding system and pollen limitation

We evaluated the breeding system and pollen limitation (PL) through manual pollination treatments. Floral buds of different individuals were previously bagged with "voile" bags and the flowers submitted to the following treatments: (1) spontaneous self-pollination–20 flowers from 8 individuals, were bagged and not manipulated in 2016; (2) hand self-pollination—49 flowers from 20 individuals, were supplemented manually with pollen from the same flower and bagged in 2016; (3) hand cross-pollination—130 flowers from 47 individuals, were supplemented with pollen from other individual, located at least 10 m away and then bagged. We conducted the cross-pollination treatment in the years 2016 (26 flowers), 2017 (48 flowers) and 2018 (56 flowers); (4) pollination under natural conditions—131 flowers from 36 individuals were marked and kept unbagged.

We evaluated the flowers from natural pollination in 2016 (49 flowers), 2017 (20 flowers) and 2018 (62 flowers). At the end of the experiments we collected the fruits and counted the number of seeds per fruit in the laboratory. We calculated the index of pollen limitation as $IPL = 1 - P_n/P_c$ where $P_n$ is the proportion of seed-bearing fruits multiplied by the mean number of seeds per fruit of flowers exposed to natural pollination and $P_c$ is the proportion of seed-bearing fruits multiplied by the mean number of seeds per fruit of flowers after hand cross-pollination (adapted from *Lloyd & Schoen (1992)*; *Larson & Barrett (2000)*). We used as response variable of reproductive success in all analyses such an estimate combining fruits bearing seeds and the number of seeds, which is appropriate because *E. lindenii* is parthenocarpic (i.e., flowers develop into fruits independent of pollination). Values of IPL ≤0.2 indicate absence of PL, whereas IPL > 0.8 indicates strong PL (*Freitas, Wolowski & Sigiliano, 2010*). We assessed the self-incompatibility by the index of incompatibility (ISI), a relative measure of the seeds produced after self- and cross-pollination (*Zapata & Arroyo, 1978*). Species with ISI < 0.30 may be classified as self-incompatible (*Ramirez & Brito, 1990*).

## Frequency and efficacy of floral visitors

We performed focal observations (sensu *Dafni, Kevan & Husband, 2005*) to evaluate the identity of floral visitors and their frequency of visits, by censuses of 30 min per individual (*n* = 190 individuals) between 6:00 AM and 12:00 PM, totalizing 184 h of observation. Observations were done in four reproductive events, in the years 2015 (43.5 h, *n* = 50 individuals), 2016 (39.0 h, *n* = 42), 2017 (51.0 h, *n* = 48), and 2018 (50.5 h, *n* = 50). Images and videos were captured during the visits to evaluate the foraging behavior and the floral resources obtained. The visits were identified as legitimate or illegitimate by the expected mode of pollination, considering the shape and arrangement of the flower parts (sensu *Irwin et al., 2010*; *Freitas, 2018*). Specimens of insects were collected for
posterior identification. We grouped the floral visitors into three functional groups based on identity, body size and foraging behavior, as following: hummingbirds, large bees (length ≥ 10 mm) and small bees (<10 mm).

We evaluated the efficacy of the three functional groups of floral visitors through experiments of selective exposition. In this experiment, flowers previously bagged at the bud stage were exposed to a single visit by any visitor of the three functional groups, as follows: small bees (48 flowers in 2016), large bees (20 flowers in 2017 and 60 in 2018) and hummingbirds (20 flowers in 2017 and 65 in 2018). Here we also considered the proportion of seed-bearing fruits multiplied by the mean number of seeds per fruit as the response variable of the pollination efficacy by each functional group.

## Data analyses

We performed all the analyses in R version 3.4.4 (*R Core Team, 2018*). We evaluated the production of nectar during anthesis and the effect of nectar removal in nectar secretion by analyses of variance (one-way ANOVA), using the function *aov*. We assessed the differences between treatments (time of anthesis and number of removals) by Tukey HSD post-hoc test, using the function *TukeyHSD*.

We conducted a linear model to evaluate if the reproductive success after natural pollination varied between three reproductive events (2016, 2017 and 2018). Prior to analyses, we cubic-root transformed values of the response variable to meet the normality assumptions. We used the reproductive events (three levels: 2016, 2017 and 2018) as fixed effect. We established the model using the function *lm* and we tested the model assumptions by visual inspection of the residuals using the *qqnorm* function. We calculated the significance of each term in the model using the function *Anova* (Type II) from the *car* package (*Kuznetsova, Brockhoff & Christensen, 2017*) and the differences between levels of categorical factors using the function *lsmeans* from *lsmeans* package (*Lenth, 2016*).

To evaluate whether hummingbirds and large bees differ in their efficacy, we conducted a linear model. Prior to analyses, we cubic-root transformed values of the response variable to meet the normality assumptions. We used the functional group of pollinators (two levels: hummingbirds and large bees) and the year when the treatments were conducted (two levels: 2017 and 2018) as fixed effects. We established the model using the function *lm* and we tested the model assumptions by visual inspection of the residuals using the *qqnorm* function. We calculated the significance of each term in the model using the function *Anova* (Type II) from the *car* package (*Kuznetsova, Brockhoff & Christensen, 2017*) and the differences between levels of categorical factors using the *lsmeans* package (*Lenth, 2016*). We did not compare the efficacy of small bees as no seeds were produced after their visits.

To evaluate if the frequency of visits varies between the four reproductive events and the group of floral visitors, we conducted a log-linear model for categorical data (*Wood, 2017*). We used the relative frequency of visits as a response variable and the reproductive events (four levels: 2015, 2016, 2017 and 2018) and the functional groups (three levels: hummingbirds, large bees and small bees) as fixed effects. We established the model using

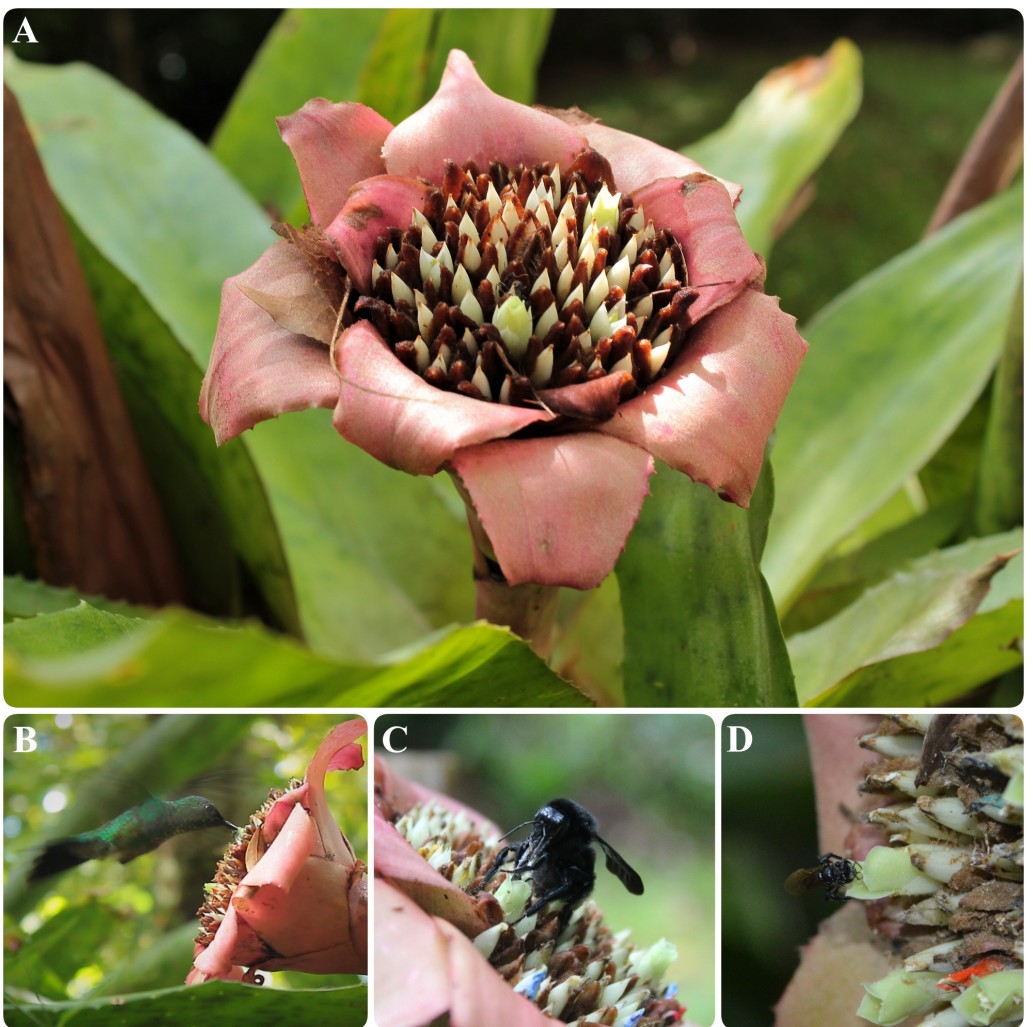

**Figure 1** *Edmundoa lindenii* **flowers were visited by three functional groups: hummingbirds, large bees and small bees.** (A) Inflorescence and flowers; (B) visit by the hummingbird *Amazilia fimbriata*; (C) visit by the large bees *Bombus morio*; (D) visit by the small bee *Trigona spinipes*. All observations were made in the montane Atlantic Forest at Serra do Órgãos National Park, southeastern Brazil.

the *glm* function with the family set to *poisson* (*Lindsey, 1997*). We tested the model assumptions by visual inspection of the residuals. We calculated the significance of each term in the model using the *anova* function (test = *Chisq*) and the differences between levels of categorical factors using the *emmeans* function available in the *emmeans* package.

## RESULTS

### Floral biology

The flowers of *E. lindenii* are grouped in a compound corymboid inflorescence with *ca*. 100–150 flowers, inserted in the leaf rosette (Fig. 1). Inflorescence diameter reached 121.32 ± 17.01 mm and scape length 296.87 ± 23.86 mm (mean ± SD throughout the text).
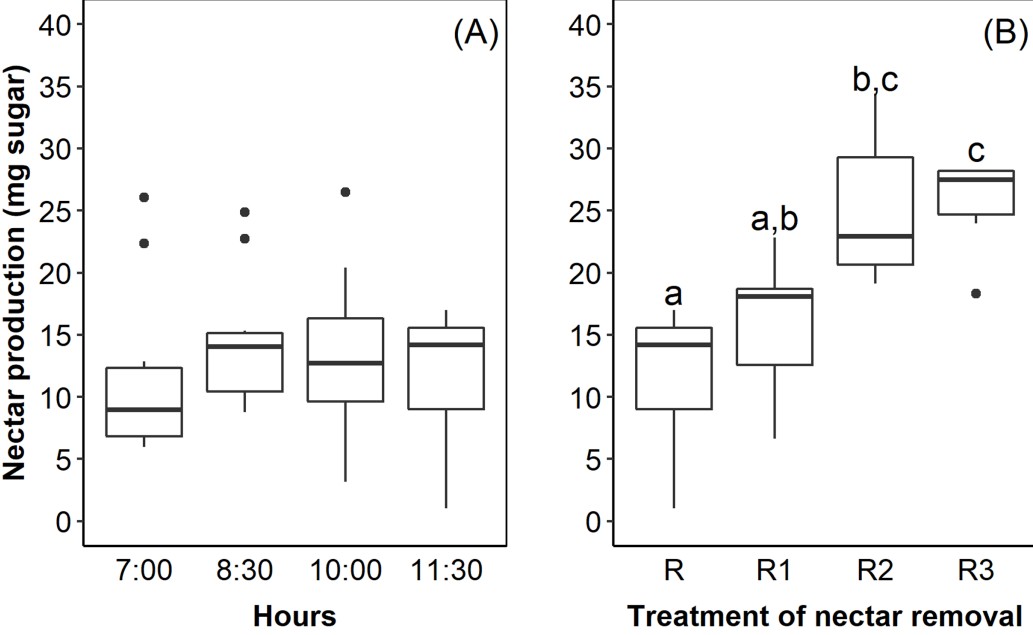

**Figure 2 Nectar production in *Edmundoa lindenii* flowers did not increase during the anthesis, but the nectar removal stimulated new secretion.** (A) Nectar production along the anthesis at four different times of the day: 7:00 AM; 8:30 AM; 10:00 AM; and 11:30 AM. (B) Nectar production after experimental removal of nectar: R, no removal; sampled one time at 11:00 AM; R1, one removal; sampled two times at 10:00 AM and 11:30 AM; R2, two removals; sampled three times at 8:30 AM, 10:00 AM and 11:30 AM; and R3, three removals; sampled four times at 7:00 AM, 8:30 AM, 10:00 AM and 11:30 AM. The horizontal line in the both boxplots represents the median values, the upper and lower sides of the box represent the corresponding quartiles and vertical lines are minimum and maximum values of the data range. Dots are outliers. Different letters indicate statistical significance between pairs of years ($p < 0.05$) by ANOVA post-hoc test (TukeyHSD).

The flowers are hermaphrodite, with the androecium presenting six stamens included in the corolla and anthers with longitudinal dehiscence (Fig. 1). The gynecium is also included in the corolla and the style ends in a three-lobed stigma (Fig. 1). The inferior and trilocular ovary contained 197.9 ± 54.12 ovules. The length of bracts and sepals was 55.15 ± 6.99 mm and 26.0 ± 4.0 mm, respectively. The corolla is tubular (length: 17.95 ± 2.92 mm) with a narrow opening (3.11 ± 1.17 mm). The flowers have diurnal anthesis, characterized by the presence of exposed pollen grains and receptive stigma. The anthesis began at around 06:00 AM and lasted for about 6 hours when corolla closed. The secretion of nectar started in the beginning of anthesis and it did not increase over time ($F = 0.44$; df = 3; $p = 0.726$; Fig. 2A). However, the removal of nectar stimulated new secretion ($F = 6.632$, df = 3, $p = 0.00273$, Fig. 2B).

The bracts reflect red wavelengths, whereas the corolla is UV-reflecting white and the sepals are UV-absorbing white (Fig. 3). The color of petals, sepals and bracts, as well as open or closed flowers, is distinguishable by bees and hummingbirds. Flower color was 1–7 times above the discrimination criteria (0.27) for bee vision (petals-sepals 4 ± 1, bracts-sepals 4 ± 2) and 5–15 times above the discrimination criteria (1.0) for hummingbirds (petals-sepals 8 ± 2, bracts-sepals 12 ± 4, bracts-petals 15 ± 7).

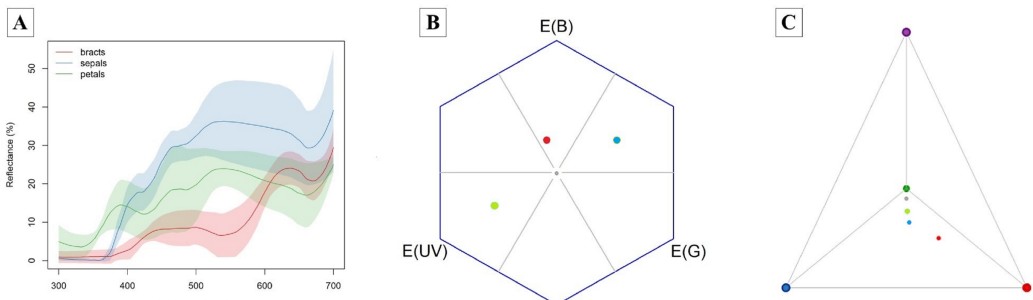

**Figure 3 Attractive structures of *Edmundoa lindenii* include red bracts, UV-reflecting white petals and UV-absorbing white sepals and can be detected by bees and hummingbirds.** (A) The reflectance spectra of attractive structures in *E. lindenii* inflorescences. For each structure, the colored line represents the mean reflectance and the corresponding color shading represents the standard deviation. Red, bract reflectance; blue, sepal reflectance; green, petal reflectance. (B) Hexagon model for bee vision based on the photoreceptors of *Bombus terrestris*. (C) Tetrahedron model for bird vision based in the photoreceptors of *Sephanoides sephaniodes*. In both models, the gray point represents achromatic center, the red point represents mean loci for bracts, the blue point indicates the mean loci for sepals, and the green point represents mean loci for petals.

**Table 1 The population of *Edmundoa lindenii* in PARNASO is self-incompatible and parthenocarpic.** Fruit and seeds production after hand pollination treatments at Serra do Órgãos National Park, southeastern Brazil. Reproductive success = proportion of fruits with seeds multiplied by the number of seeds.

| Treatments | Flowers ($n$) | Fruits with seeds ($n$) | Fruit set | Seeds (mean ± SD) | Reproductive success |
|---|---|---|---|---|---|
| Spontaneous self-pollination | 20 | 1 | 0.05 | 26 | 1.30 |
| Manual self-pollination | 49 | 4 | 0.08 | 126.00 ± 58.17 | 10.08 ± 4.65 |
| Cross-pollination total | 130 | 127 | 0.98 | 116.11 ± 58.97 | 113.64 ± 61.27 |
| Cross-pollination 2016 | 26 | 23 | 0.89 | 126.57 ± 53.66 | 99.65 ± 57.91 |
| Cross-pollination 2017 | 48 | 48 | 1.00 | 138.44 ± 63.47 | 138.44 ± 63.47 |
| Cross-pollination 2018 | 56 | 56 | 1.00 | 98.88 ± 54.65 | 98.88 ± 54.65 |
| Natural conditions total | 131 | 108 | 0.82 | 119.08 ± 66.31 | 101.98 ± 58.43 |
| Natural conditions 2016 | 49 | 45 | 0.92 | 131.19 ± 55.58 | 120.69 ± 51.13 |
| Natural conditions 2017 | 20 | 11 | 0.55 | 76.73 ± 73.66 | 42.20 ± 40.51 |
| Natural conditions 2018 | 62 | 52 | 0.84 | 118.04 ± 70.23 | 99.15 ± 58.99 |

## Breeding system and pollen limitation

*Edmundoa lindenii* is self-incompatible (ISI = 0.08; Table 1) and parthenocarpic (Table 1), requiring the action of pollinators for sexual reproduction. We observed that the reproductive success after natural pollination varied between reproductive events (ANOVA: $F = 12.9$, df = 2; $p < 0.001$). The lowest reproductive success occurred in the reproductive event of 2017 (contrasts: 2016–2017: $t = 5.050$; df = 103; $p < 0.001$; 2016–2018: $t = 2.100$; df = 103; $p = 0.095$; 2017–2018: $t = -3.840$; df = 103; $p < 0.001$; Fig. 4). Pollen limitation was expressive only in 2017 (PL index: 2016 = −0.21, 2017 = 0.70, 2018 = −0.003).
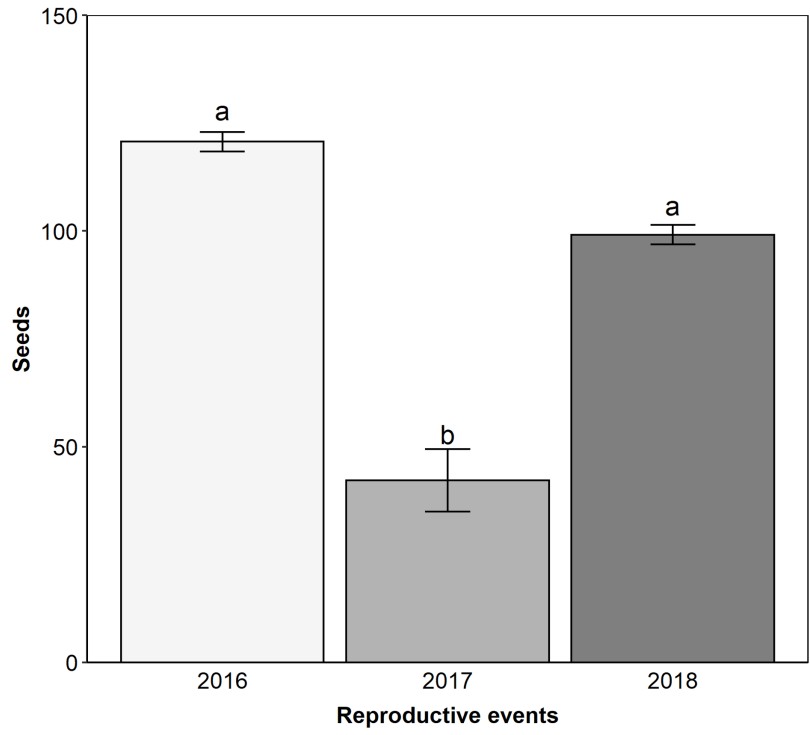

**Figure 4 The reproductive event of 2017 presented the lower reproductive success by natural pollination.** Reproductive success after natural pollination among three reproductive events (2016, 2017 and 2018). Seeds, proportion of fruits with seeds formed multiplied by the number of seeds in each fruit. Different letters show statistically significant difference ($p < 0.05$).

## Floral visitors and temporal variation

The flowers of *E. lindenii* were visited by 11 species of animals belonging to three functional groups (hummingbirds, large bees and small bees; Table 2). Hummingbirds were the group with the highest species richness with seven species (Table 2; Fig. 1). The small bee *T. spinipes* was the only visitor that conducted illegitimate visits, resulting in damage of corolla and/or anthers by chewing. Hummingbirds and large bees foraged for nectar acting as legitimate visitors and small bees collected pollen.

The pollinator functional group affected the reproductive success of *E. lindenii* (Table 3) and hummingbirds had higher efficacy than large bees (contrast: $t = 3.015$ df = 148, $p = 0.003$; Table 4; Fig. 5). Small bees did not act as pollinators, as none of the flowers they visited produced seeds. The frequency of visits varied between functional groups of floral visitors with significant interaction between functional group and reproductive event (Table 4; Fig. 6).

## DISCUSSION

The frequency of floral visitors' groups (hummingbirds, large bees and small bees) varied between reproductive events of *E. lindenii* and this variation resulted in a reduction in the natural pollination in one year, when pollen limitation was also recorded. The existence of year-to-year changes in the composition of floral visitors has been found in several

**Table 2 Floral visitors and the resources they have taken in *Edmundoa lindenii* flowers.** All records were made in 2015 to 2018 at Serra do Órgãos National Park, southeastern Brazil. Functional groups, represented by different shades: HB, hummingbirds; LB, large bees; SB, small bees. Type of resources: P, pollen; N, nectar.

| Family | Species | Functional group | Resource |
|---|---|---|---|
| Trochilidae | *Amazilia lactea* (Lesson, 1832) | HB | N |
| | *Amazilia versicolor* (Vieillot, 1818) | HB | N |
| | *Leucocholoris albicollis* (Viellot, 1818) | HB | N |
| | *Phaethornis eurynome* (Lesson, 1832) | HB | N |
| | *Ramphodon naevius* (Dumont, 1818) | HB | N |
| | *Thalurania glaucopis* (Gmelin, 1788) | HB | N |
| | *Amazilia fimbriata* (Gmelin, 1788) | HB | N |
| Apidae | *Bombus morio* (Swederus, 1787) | LB | N/P |
| | *Bombus brasiliensis* (Lepeletier, 1835) | LB | N/P |
| | *Euglossa* sp. | LB | N |
| | *Trigona spinipes* (Fabricius, 1793) | SB | P |

**Table 3 The pollination efficacy of hummingbirds was higher than efficacy of large bees in *Edmundoa lindenii*.** ANOVA results on the pollination efficacy of functional groups of pollinators of *E. lindenii*, measured by seed set after a single visit to the flower at Serra do Órgãos National Park, southeastern Brazil.

| Effects | DF | F | p |
|---|---|---|---|
| Reproductive event (year) | 2 | 0.15 | 0.860 |
| Functional group of pollinators | 1 | 71.07 | <0.001 |
| Reproductive event: functional group | 1 | 2.60 | 0.078 |

**Table 4 The reproductive events and the functional groups of flower visitors of *Edmundoa lindenii* had an effect on the relative frequency of visits.** ANOVA results on the relative frequency of floral visits by functional groups during the reproductive events of 2015, 2016, 2017 and 2018 at Serra do Órgãos National Park, southeastern Brazil.

| Effects | DF | Deviance | p (>Chi) |
|---|---|---|---|
| Reproductive events (year) | 3 | 0.020 | 0.999 |
| Functional group of floral visitors | 2 | 26.392 | <0.001 |
| Reproductive events: functional group | 6 | 244.562 | <0.001 |

systems (*Schemske & Horvitz, 1984*; *Traveset & Sáez, 1997*; *Price et al., 2005*; *Olesen et al., 2008*; *Petanidou et al., 2008*), while in others, pollination efficacy between different years was studied (*Fishbein & Venable, 1996*; *Stoepler et al., 2012*). However, there are fewer studies that consider both plant reproductive success and variations in the pollinator assemblages along time (*Herrera, 1990*; *Fleming et al., 2001*; *Salas-Arcos, Lara & Ornelas, 2017*). Thus, through observational and experimental approaches, we have shown that the

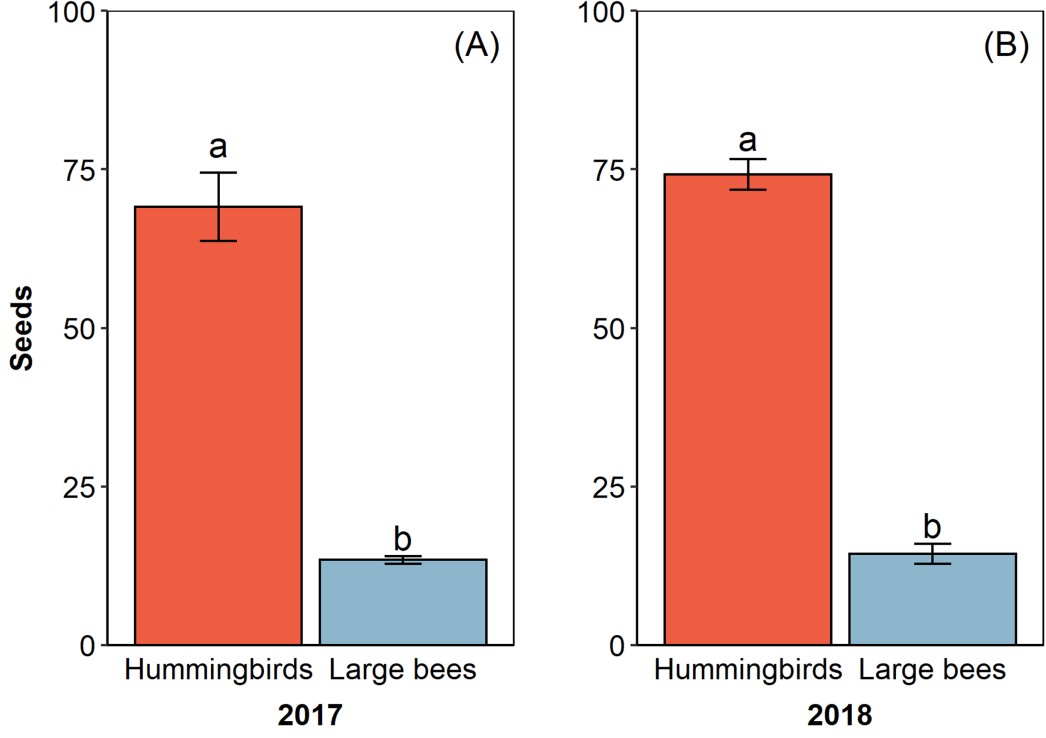

**Figure 5 Hummingbirds had higher efficacy than large bees in two reproductive events.** Pollinator efficacy in the reproductive events of 2017 (A) and 2018 (B). Seeds, proportion of fruits with seeds formed multiplied by the number of seeds in each fruit. Different letters show statistically significant difference ($p < 0.05$).

reproductive success of a generalist species responded to temporal variation of its assemblage of pollinators.

Overall hummingbirds had higher efficacy and visitation frequency than large bees, so they could be pointed as the main pollinators of *E. lindenii*. Hummingbirds also act as the main pollinators in other species of Bromeliaceae (*Schmid et al., 2011*; *Magalhães et al., 2018*). However, large bees did not always act as secondary pollinators of *E. lindenii*: they were the main pollinators when the frequency of visits by hummingbirds was low. Pollinators belonging to different functional groups may vary in their pollination effectiveness, exerting pressures toward specialization (*Stebbins, 1970*; *Rosas-Guerrero et al., 2014*). Thus, evolution of a generalized pollination system is expected when different pollinators play the same role as selective agents or if the less effective pollinators provide reproductive assurance, offsetting fluctuations of the most effective pollinators. This could be the case for large bees and *E. lindenii*, as they can ensure sexual reproduction when hummingbirds fail. In short, our results are consistent with the hypothesis that the maintenance of generalist pollination is related to the existence of variable pollination environments (*Waser et al., 1996*) and support the combined measurements of reproductive success and pollinator assemblages along time and space as an interesting approach in this regard (see *Gómez & Zamora, 2006* for additional suggestions). Curiously, there was no evidence of declining populations of hummingbirds in the study area in

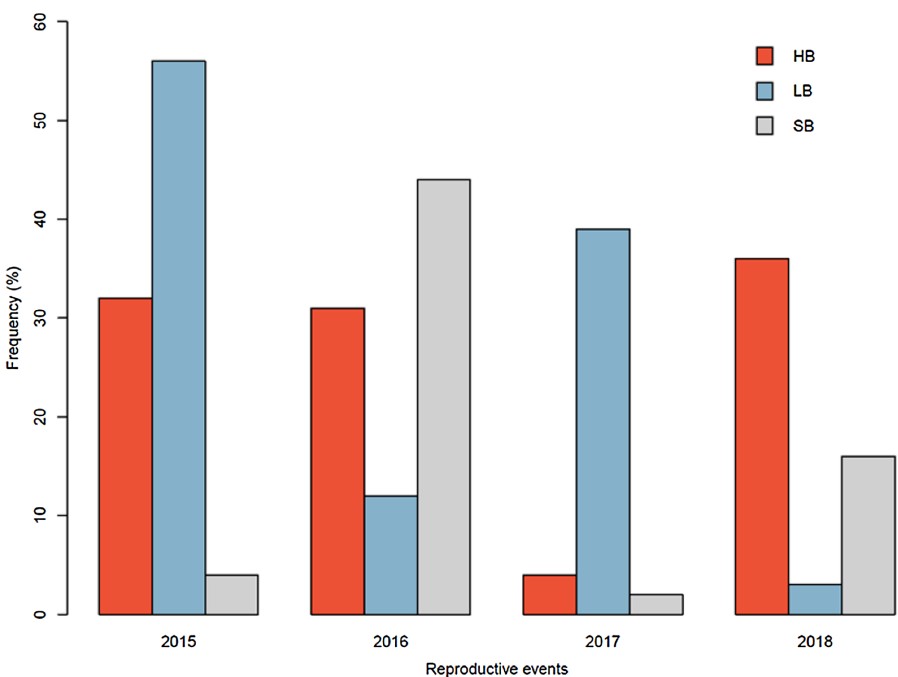

**Figure 6 Relative frequency of visits by each functional group in *Edmundoa lindenii* flowers varied among the four reproductive events.** The relative frequency of visits in the reproductive events of 2015, 2016, 2017 and 2018. HB, hummingbirds; LB, large bees; SB, small bees.

2017, the causes of the drastic reduction of visits to the flowers of *E. lindenii* by hummingbirds that year are completely unknown.

Despite the differences on effectiveness, flowers of *E. lindenii* are attractive to both hummingbirds and bees. Floral visitors identify and select flowers using a variety of characteristics, including size and color contrasts (*Papiorek et al., 2016*) and hummingbirds and bees can detect different color spectra (*Chittka & Waser, 1997*; *Vorobyev & Osorio, 1998*). The floral traits of *E. lindenii* are detectable by both groups of floral visitors. Petals had UV reflection, sepals absorbed UV and bracts were red. These results correspond to the expected pattern for attraction of bees and hummingbirds, as bees have a spectrum of vision that includes UV wavelengths, around 300–400 nm (*Kevan, Chittka & Dyer, 2001*) and hummingbirds are known for their preference for red-coloured flowers that mostly are UV-absorbent (*Lunau et al., 2011*). However, tradeoffs between selective pressures exerted by different pollinators may occur if they differ in preference for floral traits (*Gervasi & Schiestl, 2017*). As pointed out above, we may speculate that the temporal fluctuation in the visitor frequencies of *E. lindenii* may reduce the probability of pollinators exerting consistent selective pressures on its floral traits (*Schemske & Horvitz, 1984*; *Gómez & Zamora, 2006*). In this case, attractive and nonrestrictive flowers for large bees are important for reproductive assurance in reproductive events with a low frequency of hummingbirds, thus it would not be expected a shift to specialized hummingbird-flowers. However, further experiments are needed
to support that selection toward pollination specialization by hummingbirds is restrained in this system.

In addition to visual signals, the nectar of *E. lindenii* is accessible to hummingbirds and large bees, thus its flowers are attractive and legitimately accessible to the two groups. The dynamics of nectar production influences the behavior of pollinators during visits to the flowers (*Parachnowitsch, Manson & Sletvold, 2018*). Although nectar production of *E. lindenii* did not increase over time, its removal stimulated further secretion. Similar results have been found in other bromeliads, including some species visited by hummingbirds and bees (*Galetto & Bernardello, 1992*; *Ordano & Ornelas, 2004*). Nectar secretion after withdrawal may favor repeated visits to the flower. This is consistent with our results of the pollination efficacy experiment in *E. lindenii*. Specifically, seed set after one visit by a large bee or hummingbird was lower than seed set of flowers exposed to pollinators during the whole anthesis, indicating that more than one visit is necessary to achieve maximum fecundity in this species. At last, nectar secretion after removals and repeated visits by pollinators could be related to the male component of the reproductive success (*Ordano & Ornelas, 2004*).

The population of *E. lindenii* at PARNASO was self-incompatible, but self-compatibility has been registered in other populations of this species (*Matallana et al., 2010*). Variations between self-incompatibility and self-compatibility within species are common in plant evolution and may indicate transitions between reproductive systems (*Igic, Lande & Kohn, 2008*; *Moreira, Miranda & de Lima, 2017*). Studies have shown that compatibility barriers can be broken by genetic changes (such as mutations) (*Sassa et al., 1997*), physiological factors, elevated temperatures and stress (*Tezuka et al., 1997*), allowing for self-pollination. Moreover, breeding systems may be related to the degree of pollination generalization, linking shifts in pollination and incompatibility systems. For instance, *Wessinger & Kelly (2018)* found a relationship between self-compatibility and attributes related to the attraction of hummingbirds, including red flowers and loss of floral aroma and UV-absorbing pigments. In contrast, we found self-incompatibility and UV-absorbing sepals in *E. lindenii* despite higher frequency of visits by hummingbirds. Self-incompatibility in this species seems to work as a barrier to autogamous pollination, since small bees access the anthers, make long visits to the flower, and manipulate the pollen. In fact, pollinators usually do not operate independently of herbivores (florivores in this case), which may generate a tradeoff between the fitness functions by each kind of organism (*Ashman, 2002*; *Gómez & Zamora, 2006*; *Gélvez-Zúñiga et al., 2018*).

## CONCLUSION

Our results allow us to conclude that hummingbirds and large bees were the main and the secondary pollinators of *E. lindenii*, respectively. Moreover, small bees had a negative effect on its reproduction. These results could indicate a higher degree of specialization of this system than the apparent generalization considering floral visitor composition (*Padyšáková et al., 2013*), which would be in accordance with the most effective pollinator principle (*Stebbins, 1970*). However, temporal variations in the pollinator assemblages had effects on reproductive success of *E. lindenii*, leading to the occurrence of pollen

limitation when visits by hummingbirds were scarce. Despite their lower pollination efficacy, large bees ensured seed set when hummingbirds failed. Plant pollination generalization has been associated with similar effectiveness by the different pollinators (*Waser et al., 1996*). However, we provide evidence that variable pollination environments may favor generalization, even under differential effectiveness of pollinator groups if secondary pollinators provide reproductive assurance. This reinforces the idea of different mechanisms driving the evolution of generalized pollination systems (*Schiestl, Balmer & Gervasi, 2018*).

## ACKNOWLEDGEMENTS

The authors thank Gabriel C. Rocha for field and data analyzing assistance; Cristovão Albuquerque (Publicase, Brazil) for language editing; Mariano Ordano for critical review; and the staff of the Serra dos Órgãos National Park for logistics support and for allowing access to Park facilities.

### Funding

The National Council for Scientific and Technological Development (CNPq), Brazil provided support for the project through funding and in the form of undergraduate scholarships (PIBIC) to Julia de Oliveira Ferreira and Roberta Luísa Barbosa Leal and researcher scholarships to Leandro Freitas (PQ) and Miguel Angel Rodríguez-Gironés (PVE). The Coordination for the Improvement of Higher Education Personnel (CAPES), Brazil provided graduate scholarships to Alessandra Ribeiro Pinto and Marina Muniz Moreira and partial funding (Finance Code 001). The Foundation for Research Support of the State of Rio de Janeiro (FAPERJ), Brazil provided researcher scholarship to Leandro Freitas (CNE). The funders had no role in study design, data collection and analysis, decision to publish, or preparation of the manuscript.

### Grant Disclosures

The following grant information was disclosed by the authors:
The National Council for Scientific and Technological Development (CNPq), Brazil.
The Coordination for the Improvement of Higher Education Personnel (CAPES), Brazil, Finance Code: 001.
The Foundation for Research Support of the State of Rio de Janeiro (FAPERJ), Brazil.

### Competing Interests

The authors declare that they have no competing interests.

### Author Contributions

- Roberta Luisa Barbosa Leal conceived and designed the experiments, performed the experiments, analyzed the data, prepared figures and/or tables, authored or reviewed drafts of the paper, and approved the final draft.

- Marina Muniz Moreira conceived and designed the experiments, performed the experiments, analyzed the data, prepared figures and/or tables, authored or reviewed drafts of the paper, and approved the final draft.
- Alessandra Ribeiro Pinto performed the experiments, analyzed the data, prepared figures and/or tables, authored or reviewed drafts of the paper, and approved the final draft.
- Júlia de Oliveira Ferreira performed the experiments, authored or reviewed drafts of the paper, and approved the final draft.
- Miguel Rodriguez-Girones conceived and designed the experiments, analyzed the data, authored or reviewed drafts of the paper, and approved the final draft.
- Leandro Freitas conceived and designed the experiments, prepared figures and/or tables, authored or reviewed drafts of the paper, and approved the final draft.

## Field Study Permissions

The following information was supplied relating to field study approvals (i.e., approving body and any reference numbers):

The field research reported here was performed using the required permit at SISBIO (SISBIO Nos. 34882 and 432793).

## Data Availability

Raw data is available as Supplemental Files.

## Supplemental Information

Supplemental information for this article can be found online at http://dx.doi.org/10.7717/peerj.8836#supplemental-information.

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
