# Peer review of "Temporal changes in the most effective pollinator of a bromeliad pollinated by bees and hummingbirds"

_PeerJ, doi:10.7717/peerj.8836_

## Round 0.1 · original submission · Major Revisions

Hi there,
Apologies for the delay with this manuscript. Please address the comments stated by the reviewers and re-submit for reconsideration. Thank you

Reviewer 1 ·

Basic reporting

BASIC REPORTING
Clear, unambiguous, professional English language used throughout.

Line 31: Note that the use of “efficacy” “efficacious” (e.g., line 288 and elsewhere) might be confusing for the general reader. Please consider explaining somewhere in the MS the difference between most efficient pollinator, most effective pollinator and most efficacious pollinator. Also, the use of “intermediary” (line 52) might be misleading, try “intermediate”. In general, the text is clear, unambiguous, and professional English language is used throughout.

Intro & background to show context. Literature well referenced & relevant.

Structure conforms to PeerJ standards, discipline norm, or improved for clarity.

Figures are relevant, high quality, well labelled & described. Yes.

Raw data supplied (see PeerJ policy). 26]. All data generated in this study have been presented. Though morphological data should be publicly available, perhaps at Dryad or as supplementary in PeerJ? Some of the videos of floral visitors would be neat to see at PeerJ.

Experimental design

EXPERIMENTAL DESIGN
Original primary research within Scope of the journal.
Research question well defined, relevant & meaningful. It is stated how the research fills an identified knowledge gap.
Rigorous investigation performed to a high technical & ethical standard.
Methods described with sufficient detail & information to replicate.

Line 147 and elsewhere. Please explain if flowers last one day.

Table 2. Based on the list of floral visitors, it would be interesting to see whether all hummingbird species are similarly effective pollinators and abundant over time.

Validity of the findings

VALIDITY OF THE FINDINGS
Impact and novelty not assessed. Negative/inconclusive results accepted. Meaningful replication encouraged where rationale & benefit to literature is clearly stated.
Data is robust, statistically sound, & controlled.
Conclusions are well stated, linked to original research question & limited to supporting results.

Lines 282–297. The occurrence of PL was recorded in E. lindenii only in the year in which there was a strong decrease in the frequency of visits by hummingbirds, when large bees were the most frequent visitors. Thus, the effectiveness of large bees alone was not enough to achieve the potential seed set fully. This is correlative. An experimental approach might be needed to tease apart the causal factors incl. an experiment in which flowers are pollinated by both functional groups.

Lines 290–292. Lastly, combined measurements of PL and pollinator assemblages along time and space is an interesting approach to evaluate the effects of variable pollination environments (see Gómez & Zamora, 2006 for other suggestions in this regard). Indeed. Hard work is needed to get both temporal and spatial variation of pollinator environments in plants with generalized-pollination systems and parthenocarpic plants as the one studied. Please explain the implications of these in the Discussion.

Line 307 and elsewhere. It might be useful to discuss further the meaning of the positive response to nectar removal in the studied species in terms of its pollination dynamics and what has been found in other species of bromeliads (e.g., Ordano & Ornelas 2004, Oecologia).

Lines 318–324. Wessinger and Kelly (2018) found a relationship between self-compatibility and attributes related to the attraction of hummingbirds, including red flowers and loss of floral aroma and UV-absorbing pigments. Most hummingbird-pollinated species are self-compatible (Wessinger and Kelly 2018; see also Raduski et al. 2012 Evolution, Wolowski et al. 2013 The Science of Nature). If the studied plant species is more specialized on hummingbird pollination, bird-flower traits might be selected with increased hummingbird visitation over evolutionary time, resulting in specialization to (e.g., replenishment of removed nectar, large amounts of nectar) and the evolution of floral traits present in hummingbird-pollinated species (Wessinger and Kelly 2018, Salas-Arcos et al. 2019 The Science of Nature). It seems that your data points toward that direction.

Lines 329–331. Do you have data on pollen transfer? Are hummingbirds most effective in pollen transfer among plants?

Speculation is welcome, but should be identified as such.

Additional comments

Review of PeerJ-2019-35011

Roberta L B Leal, Marina M Moreira, Alessandra R Pinto, Julia O Ferreira, Miguel Rodriguez-Girones & Leandro Freitas

“Temporal changes in the most effective pollinator on a bimodal system involving bees and hummingbirds”

The authors evaluated the pollination effectiveness of the floral visitors of Edmundoa lindenii (Bromeliaceae) across four reproductive events. They analyzed the frequency of floral visitors (large bees, small bees, and hummingbirds) through focal observations and their single-visit efficacy (seed set). Pollen limitation (PL) index was estimated comparing seed set after hand cross and natural pollination. Edmundoa lindenii is self-incompatible and parthenocarpic, requiring the action of pollinators for reproduction. Hummingbirds have greater efficacy than large bees, and small bees act as pollen robbers. The frequency of floral visitors varied among years, and overall hummingbirds were more effective than large bees. The PL index also varied over time, with limitation only occurring in the reproductive event of 2017, when hummingbirds were scarce. The authors conclude that a generalist species can suffer or not PL in different reproductive events, in response to variations in the pollinator assemblage. Although the evolution of a generalized pollination system is expected when different pollinators play the same role as selective agents, their results support the hypothesis that generalization may also be favored when less effective pollinators provide reproductive assurance, lightening fluctuations of the most effective pollinators, such as could be the case for large bees and E. lindenii. Their conclusions are well stated, linked to original research question and limited to supporting results. However, there are few important issues (mainly in the Discussion section) that must be considered before this manuscript is recommended for publication in PeerJ. I offer the following broad areas of criticism and suggestions for improvement:

·

Basic reporting

Writing in general is good. However, as a non-native writer, I suggest a deeper proof reading by a native writer. However, I see the major problem in the conceptual grounds. Albeit the study have not deep complex matters, the approach is in some sense a list of good measurements on the reproductive biology of an interesting system. Thus, the link between the pollination modes and the factors explaining the variation in phenotypic traits involved in the interaction and in the plant reproductive success, would be articulated in a deeper way,
In one way, the study rely on the reproductive biology of an interesting system, and less in the drivers of pollination limitation and their relationship with pollinator functional groups.
I am some uncomfortable, because the idea sounds good, but in each section the manuscript have presentation problems. This appears as a study case of reproductive biology, more than the title and abstract promet to readers.

In other aspects, I agree with data sharing. However, lack codes of sheets and variables in the datasheet. Also, revise English, there are minor finger mistakes.
Following with this transparency guidelines, it would be fine see our studies meeting the three basic rules of modern reporting, repeatability, reproducibility, and verifiability. However, this study fails respect to these points. Some points of methods must be revised and reported. Data analysis is foggy, and together data, you would be show R scripts. I found here some point for reviewing or for deeper explanations about why and how you did that you did.

Figures may be improved. Moreover if you are an R user.

Experimental design

The sampling protocol in the field is in general good. And by my experience, is generally difficult get a lot of data from this type of systems. However, several points in the methods section require a clearer presentation.

Validity of the findings

I search several times what is the novelty of this study. I find the system very interesting, also with "appeal". However, the conceptual aspects are entangled in the introduction and in the discussion. Moreover, some results referred as conclusions, have not their corresponding statistical table (for example, models). Because hypothesis and predictions are lost in the sea of a good list of traits and matters of the system, then the conclusions appear poorly linked to those conceptual grounds.

Additional comments

This is a pretty system. I encourage you to do a deeper work for improve the presentation and clarity of the idea that you wish to communicate.

---

## Round 0.2 · Minor Revisions

Hi there,

Please address the few additional comments from this reviewer. I look forward to your re-submission.

·

Basic reporting

The new version of the manuscript is better than before.

Experimental design

The data analysis is confusing.

The analysis of variance is a case of linear model. Then, you should re-write the sentences where you refer to ANOVA and linear modelling.

If you apply glm to the frequency, then you might to do the same to other response variables. Thus, you may preclude the ANOVA and apply the same approach (glm or lmer) for all cases, with differences in the error structure (Normal, Poisson, others).

On the other hand, modelling is also confusing. For example, year is a random factor when you analyze the success, but year is a fixed factor when you analyze the frequency.

You must be congruent to analyze the response variables. Then, you should take care for writing in a conceptually correct way.

Validity of the findings

In the new version of the manuscript, discussion and conclusions are better stated than before.
I found a bit of noise in the fact that the findings are not completely articulated. For instance, at the conclusions the authors wrote: "The factors that influenced the temporal variation in pollinator frequencies in our system of study are not well understood and cannot yet be predicted." The year-to-year variation is treated as a "black-box", with unmeasured factors possibly driving this variation. Perhaps the main point of the study is represented by the differences between bees and humingbirds, and the functional roles of the different groups.

Additional comments

The presentation of the study is clearly better than before. However, I found some noises yet. You should improve the modelling and/or their corresponding explanations in the Data Analysis section.
On the other hand, in conceptual terms, the relationship between the differences among different types of pollinators and the temporal variation need a deeper work.

---

## Round 0.3 · accepted · Accept

Thank you for addressing the reviewer comments.